# Towards an interoperable perovskite description or how to keep track of 300 perovskite ions

Ayman Maqsood [1,2], Hampus Nässström [3], Chen Chen [2], Li Qiutong[2], Jingshan Luo [2], Rayan Chakraborty [4], Volker Blum [4], Eva Unger [5,6], Claudia Draxl [3], José A. Márquez [3] ✉ & T. Jesper Jacobsson [7] ✉

Hybrid perovskites are interesting optoelectronic materials. The perovskite $ABX_3$ structure offers a vast compositional space, and we have identified over 300 perovskite ions. This flexibility enables tuneable properties and has significantly contributed to the success of perovskite optoelectronics. However, this diversity also leads to confusion, ambiguity, and inconsistencies causing challenges for data mining and machine learning applications. To address this issue, we propose guidelines and a JSON schema to standardize the reporting of perovskite compositions. The schema adheres to IUPAC recommendations and is designed to make data both human- and machine-readable. It captures key descriptors such as perovskite composition, molecular formula, SMILES representation, IUPAC name, and CAS number for each ion. To facilitate adoption, we have developed utilities to automatically generate comprehensive and standardized perovskite descriptions from standard ion abbreviations and stoichiometric coefficients. Additionally, we provide a curated database of all identified perovskite ions with associated descriptive data.

Understanding a phenomenon requires a precise linguistic framework that distinguishes subtle variations between seemingly similar entities. The essence of effective scientific communication lies in the collective construction of such a specialized language. This paper focuses on hybrid metal halide perovskites, highlights common ambiguities in their descriptions, and proposes a simple, interoperable format for a standardized representation of hybrid perovskite compositions.

Hybrid metal halide perovskites have gained prominence as photoabsorbers in solar cells, as evidenced by over 35,000 published papers on the subject. This surge of interest has resulted in record solar cell efficiencies exceeding 26%[1,2], with perovskite tandem cells emerging as a potentially competitive technology[3,4]. Beyond solar cells, potential applications of hybrid perovskites include LEDs[5,6], lasers[7], photodetectors[8,9], and X-ray detectors[10].

An important factor behind the success of perovskite optoelectronics is that they have a large compositional flexibility, which enables precise tuning of material properties such as the band gap[11,12]. However, this compositional diversity has also led to inconsistent nomenclature resulting in ambiguities throughout the literature. It may seem simple to write a perovskite's formula in a clear and consistent manner. After all, a perovskite has a well-defined $ABX_3$ structure (Fig. 1a), and there are IUPAC naming conventions to rely on[13]. The

[1]Helmholtz-Zentrum Berlin für Materialien und Energie (HZB), Competence Centre Photovoltaics (PVcomB), Schwarzschildstraße, 12489 Berlin, Germany. [2]Institute of Photoelectronic Thin Film Devices and Technology, State Key Laboratory of Photovoltaic Materials and Cells, Tianjin Key Laboratory of Efficient Utilization of Solar Energy, Ministry of Education Engineering Research Center of Thin Film Photoelectronic Technology. Nankai University, Tianjin, China. [3]Physics Department and CSMB Adlershof, Humboldt-Universität zu Berlin, Berlin, Germany. [4]Thomas Lord Department of Mechanical Engineering and Materials Science, Duke University, Durham, NC, USA. [5]Helmholtz-Zentrum Berlin für Materialien und Energie GmbH, HySPRINT Innovation Lab: Hybrid Materials Formation and Scaling, Berlin, Germany. [6]Division of Chemical Physics and Nano Lund, Lund University, Lund, Sweden. [7]Department of Physics, Chemistry and Biology (IFM), Linköping University, Linköping, Sweden. ✉e-mail: jose.marquez@physik.hu-berlin.de; Jacobsson.jesper.work@gmail.com

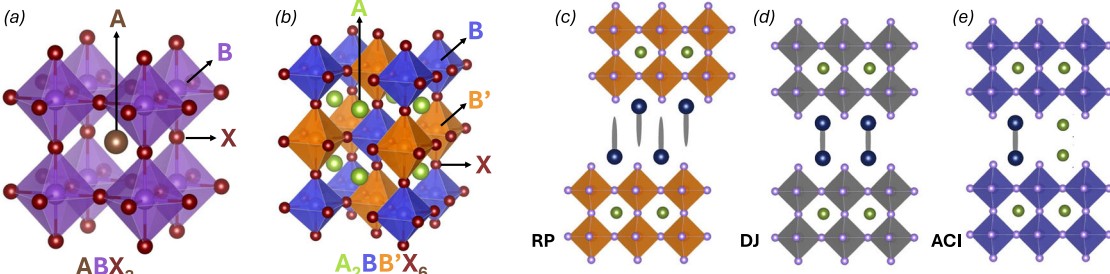

**Fig. 1 | Perovskite and perovskite-like structures. a** Idealised cubic structure for an $ABX_3$ perovskite. The B-site cations are surrounded by six X-site anions forming an octahedra. Those octahedra form a corner sharing network, and the cuboctahedral space between those octahedra is filled by the larger monovalent A-site cation. **b** An $A_2B(I)B(III)X_6$ double perovskite where the A-site cations are surrounded by a network of alternating $BX_6$ and $B'X_6$ octahedra. **c–e** 2D-perovskite structures. **c** An $A'_2A_{n-1}B_nX_{3n+1}$ Ruddlesden-Popper (RP) perovskite, **d** An $A'A_{n-1}B_nX_{3n+1}$ Dion-Jacobson (DJ) perovskite and, **e** An $A'A_nB_nX_{3n+1}$ alternating cation interlayer (ACI) perovskite. Images generated with Vesta[60].

recommendation is to list the A-site cations first, followed by the B-site cations, and then the X-site anions, sorting ions within each group alphabetically, and specifying their stoichiometric coefficients. Despite these guidelines, the perovskite literature remains highly inconsistent, and in terms of standards for interoperable nomenclature, it is a collective failure. Consider for example the perovskite composition $Cs_{0.05}FA_{0.81}MA_{0.14}PbBr_{0.45}I_{2.55}$, common in photovoltaic studies[14–16], where MA and FA are abbreviations for methylammonium and formamidinium. Numerous variations exist in how this composition has been reported. Some of the most common inconsistencies include altering the ion order, adding unnecessary parentheses, incorporating elements not present in the structure, or misrepresenting the compound as a mixture of two different phases (see Table 1).

For experienced readers, this variability in how perovskite compositions are represented is usually not a problem when referring to standard formulations. With some domain knowledge, the intended composition is often clear. However, for newcomers and less experienced readers, these inconsistencies present a challenge. The composition text string itself does not inherently clarify which ions that occupy the A, B, or X sites, and for organic A-site cations, the acronyms used in formulas can be both non-intuitive and ambiguous.

The challenge of decoding composition strings becomes even more pronounced when the reader is a machine. Historically, that has seldom been the case, but with the increasing accessibility and popularity of machine learning and data mining tools, computational readability has become an important concern. There is also a growing awareness of the value of FAIR data principles, which emphasize that data should be Findable, Accessible, Interoperable, and Reusable[17,18]. However, the inconsistent representation of perovskite compositions in standard text formats falls short of these principles. For a machine, it is not inherently clear that the different text strings in Table 1 refer to the same composition. While detailed parsers could be developed to account for inconsistencies, human ingenuity in creating new variations frequently introduces new edge cases, making such solutions brittle and error-prone.

These challenges are further amplified if also considering double perovskites and two-dimensional (2D) perovskites. Double perovskites, with the general formula $A_2B(I)B(III)X_6$, have the A-site cations surrounded by an alternating network of $BX_6$ and $B'X_6$ octahedra (Fig. 1b). 2D-perovskites consist of layered perovskite phases separated by large A-site cations that are too bulky to fit into the cuboctahedral voids between the corner sharing $BX_6$ octahedra in the 3D structure. The reduced size constraints on the A-cations results in a vast compositional diversity[19–21], with at least three structural types: Ruddlesden-Popper (RP) ($A'_2A_{n-1}B_nX_{3n+1}$), Dion-Jacobson (DJ) ($A'A_{n-1}B_nX_{3n+1}$), and alternating cation interlayer (ACI) perovskites ($A'A_nB_nX_{3n+1}$) (Fig. 1c–e). The perovskite

dimensionality can be reduced even further by incorporating even larger or branched A-site cations, or B-site cations with a valency greater than two. This could result in 1D chains of $BX_6$ octahedra, or even fully isolated $BX_6$ octahedra[22]. The latter scenario also includes vacancy-ordered perovskites, represented with the general formula $A_2V'(IV)X_6$ where V' represents a vacant site[23]. Strictly speaking, only the $ABX_3$ structure in Fig. 1a represents a perovskite structure, but in practice, both 2D-perovskites and double perovskites are widely considered part of the hybrid perovskite family.

To fully leverage the power of data mining and machine learning to uncover insights hidden in databases and literature[24], a more standardized and interoperable format for representing perovskite compositions is needed. One option to consider is the Crystallographic Information File (CIF) format, widely used in computational materials science and crystallography for defining atomic coordinates in crystal structures[25]. However, while suitable for describing crystal structures, the CIF format does not address the needs of the experimentalists generating much of the hybrid perovskite literature. An experimentalist who has spin-coated a perovskite film from a liquid solution in a less-than-pristine glove box typically does not have access to the atomic coordinates. What common experimental procedures provide is instead an educated estimate of the likely composition of the deposited perovskite based on precursor concentrations, optical measurements, and XRD-data. It is this estimate we need a user-friendly interoperable format to communicate.

In this paper, we introduce a consistent and interoperable method for representing perovskite compositions using a data model following the JSON Schema specifications[26,27]. We also provide online tools and a Python-based parser for generating this type of perovskite representation, along with an overview of the diversity of perovskite compositions found in the literature.

## Results and discussion

### Defining the composition through ions and stoichiometric coefficients

Accurately describing a perovskite composition requires a precise identification of the ions occupying the A, B, and X sites, along with their stoichiometric coefficients. Furthermore, to ensure consistency, reproducibility, and computational accessibility, this information must be represented in a structured, machine-readable format. While this goes a long way towards a proper description, the perovskite community still needs better guidelines and tools to consistently report unambiguous perovskite compositions.

This initiative is born out of the Perovskite Database Project[15,16], which compiled device data for over 40,000 perovskite solar cells from the published literature. One of the realizations from that undertaking was that perovskite compositions reported as free text suffer from extreme variability, making it difficult to maintain

**Table 1 | Examples of how the same perovskite composition can be written in different ways**

| Structure representation | Comment |
|---|---|
| $Cs_{0.05}FA_{0.81}MA_{0.14}PbBr_{0.45}I_{2.55}$ | Correct |
| $Cs_{0.05}FA_{0.81}MA_{0.14}PbI_{2.55}Br_{0.45}$ | X-ions in the wrong order |
| $Cs_{0.05}MA_{0.14}FA_{0.81}PbBr_{0.45}I_{2.55}$ | A-ions in the wrong order |
| $Cs_{0.05}(FA_{0.85}MA_{0.15})_{0.95}PbBr_{0.45}I_{2.55}$ | Unnecessary bracket on the A-side |
| $Cs_{0.05}FA_{0.81}MA_{0.14}Pb(Br_{0.15}I_{0.85})_3$ | Unnecessary bracket on the X-side |
| $(Cs_{0.05}FA_{0.95}PbBr_{0.45}I_{2.55})_{0.85}(MAPbBr_{0.45}I_{2.55})_{0.15}$ | Unreasonable indication of amalgamation |
| $(CsPbI_3)_{0.05}(FAPbI_3)_{0.81}(MAPbBr_3)_{0.14}$ | Unreasonable indication of amalgamation |
| $Cs_{0.05}FA_{0.81}MA_{0.14}PbBr_{0.45}Cl_xI_{2.55}$ | Inclusion of something that is not in the structure |
| Combinations of all the above | Leads to a combinatorial number of variations |
| – | – |

This represents a small sample of the format variations found in the literature as each example in the table only features one type of error.

formatting consistency. The variations and ambiguities were, in fact, overwhelmingly large. To address this issue, we adopted a structured approach that involved: 1) listing the A-site ions in alphabetical order, 2) listing the stoichiometric coefficients for the A-site ions, 3) listing the B-site ions in alphabetical order, 4) listing the stoichiometric coefficients for the B-site ions, 5) listing the X-site ions in alphabetical order, and 6) listing the stoichiometric coefficients for the X-site ions. This required six separate entries, but we found it necessary to achieve consistency. Once the individual ions and their coefficients are provided, automated normalization routines can be applied, complementary data can be retrieved, and a comprehensive, standardized description of the perovskite composition can be generated.

## An unambiguous description of ions

A frequently overlooked aspect of perovskite formulas is the precise identification of the ions within the structure. Elemental ions are inherently well-defined and uniquely specified by their elemental symbols, e.g. $Pb^{2+}$, $Sn^{2+}$, $I^-$, $Br^-$, etc. However, for molecular ions, the situation is more ambiguous. These ions are typically represented by 2–5 letter abbreviations, e.g., MA, FA, etc. To know that MA and FA represent methylammonium and formamidinium only requires a minimum of domain knowledge. However, the vast compositional landscape of hybrid perovskites – especially when factoring in 2D and double perovskites – necessitates a more systematic and standardised notation.

Concerning the B-site ions. The vast majority of studies focus on $Pb^{2+}$ (around 98%), $Sn^{2+}$, or a combination of $Pb^{+2}$ and $Sn^{2+}$[15]. Besides these, around 30 additional elemental ions have been explored[15], typically either with limited success as the primary B-site occupant or in minor proportions alongside $Pb$[2].

There is less variability on the X-site. There we find halides (i.e., $I^-$, $Br^-$, $Cl^-$, $F^-$), a few pseudo halides such as $CNS^-$, $F_6P^-$, and $BF_4^-$, and occasionally $O^{2-}$ or $S^{2-}$. In over 99% of all reports, the X-ions are either $I^-$, $Br^-$, or a combination of $I^-$ and $Br^-$[15].

The largest variability, and source of confusion, is found on the A-site. The Perovskite Database does for example contain 94 unique A-site ions[15]. The 2D-perovskite database[19], which also includes non-device data, contains an additional 177 A-site ions. The catalog of available perovskite precursors from Great Cell Solar adds an additional 22. In total, we have identified 293 A-site cations used for making hybrid perovskites. As the exploration of new perovskites is an ongoing endeavour, this list will most likely be incomplete. The close to 300 identified A-site ions, along with their somewhat arbitrary abbreviations, are depicted in Fig. 2.

While a 2–5 letter abbreviation for the ionic name simplifies everyday communication, it also becomes a source of confusion when the number of ions is large. Some ions have been represented by multiple abbreviations, and conversely, some abbreviations have been used for multiple ions. Another problem with short abbreviations is that they do not specify the structure or the charge of the ion. In journal articles, surprisingly often ion abbreviations are stated without any further reference, leaving the reader with the need to infer the exact nature of the ions used, which can be quite difficult.

To address these issues, we propose that when reporting a perovskite composition, the description should for each ion explicitly include: a) the canonical SMILES string, for computational representation, b) the systematic IUPAC name, for unambiguous naming, and c) the CAS-number (when available), for cross-referencing against chemical databases. We also propose to include the common name and the molecular formula to improve human readability.

This comprehensive approach ensures a clear and distinct description of perovskite compositions. Additionally, with SMILES representations, the ions can be directly processed by chemical computation and visualisation tools. It also enables the computation of molecular descriptors and molecular fingerprints, using software tools such as RDKit[28], Mordred[29], or any of a large number of alternatives, which can be used as features in machine learning models.

A problem with CAS numbers is that they are not yet defined for all ions. A practical workaround is to also provide data for a neutral parent compound, for which CAS numbers are more commonly defined. Also including the SMILES string and the systematic IUPAC name for the parent compound deals with the ambiguity of the charge for diamines, which can be either +1 or +2. Furthermore, this provides an entry point to what often is a commercially available starting material. For tertiary amines, there are no unique parent compounds, but the I, Br, or Cl salts can take that role.

Including all this information provides a somewhat lengthy perovskite description. To simplify the generation of such descriptions, we have compiled a dataset of the identified perovskite ions that includes the additional data described above. The data for each ion have been checked against the PubChem database[30]. The ions and the associated data are provided in the form of an Excel-file in the supporting information, integrated in our provided utilities, and accessible in a public database[31] hosted on NOMAD (See code and data availability for links).

## The hybrid perovskite ions database

The Hybrid Perovskite Ions Database in NOMAD provides a dedicated search interface and full programmatic access via the NOMAD API[31]. This enables searching, filtering, and visualization of perovskite ion data (Fig. 3a–c). The database also includes ion conformers calculated using RDKit, which enhance the dataset's practical utility and facilitate the development of additional applications.

The database serves as a valuable resource for experimentalists looking to identify ions for the synthesis of new hybrid perovskites,

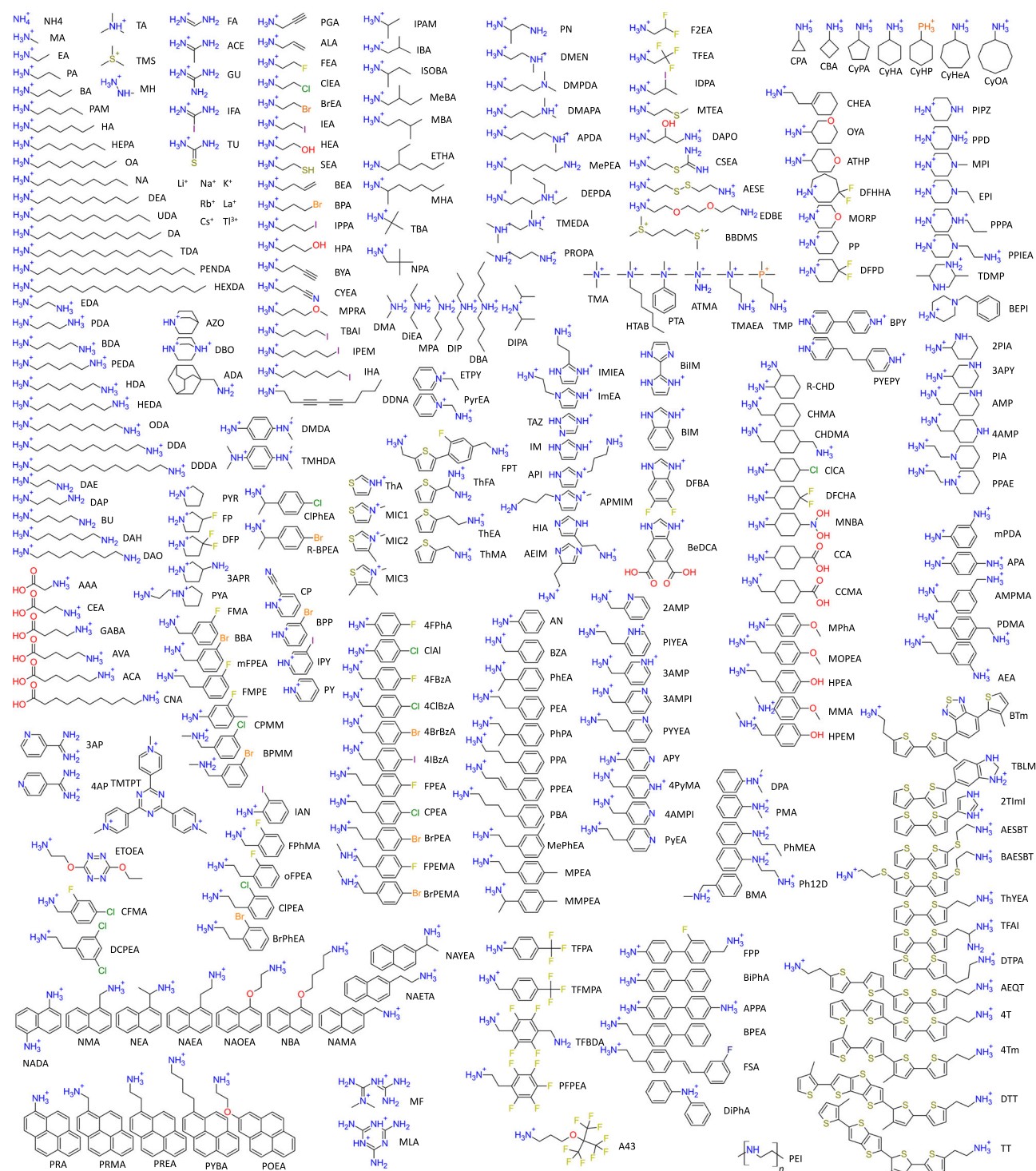

**Fig. 2 | A-site cations. Illustration of all A-site cations used to form perovskites mentioned in the Perovskite Database,1[5] the 2D-perovskite database,1[9] and the catalogue of Great Cell Solar.** The abbreviations are somewhat arbitrary, which is unavoidable. The abbreviation, molecular formula, SMILES string, IUPAC name, and CAS number (when available) for all ions are listed in the NOMAD dataset as well as in an Excel file found in both the SI and the associated GitHub repository. Data for the uncharged parent compound are listed there as well.

trace parent compounds, and select appropriate chemical precursors. With search and filtering capabilities based on descriptors such as atomic composition and molecular mass, it is, for example, possible to pinpoint larger A-site cations – which can be used to design novel compositions of 2D perovskites, such as Dion–Jacobson and Ruddlesden–Popper structures. This is an example of an interesting application given the success of 2D/3D heterostructures where lower-dimensional capping layers composed of bulkier A-site cations have resulted in improved device performance[32,33].

The Hybrid Perovskite Ions Database is also a valuable resource for computational materials scientists. It can, for example, be used to construct new systems for ab initio calculations; or be integrated directly into emerging machine-learning potential models to improve structure relaxation and property prediction. An example of a

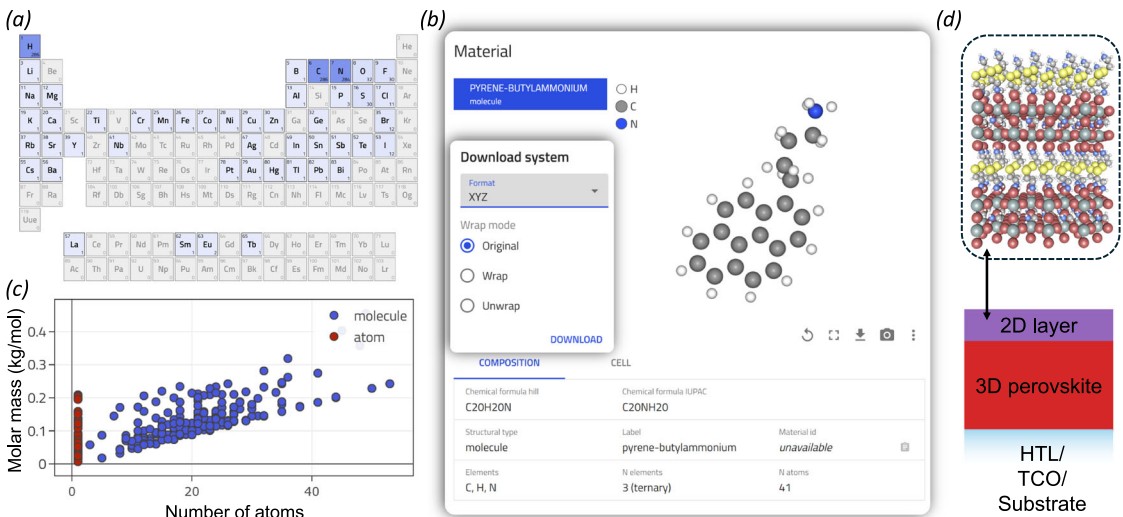

**Fig. 3 | The perovskite ion database in NOMAD. a** Heatmap showing the elements included in the database, with colour intensity representing their relative frequency. **b** Example of a material card for Pyrene-butylammonium, highlighting its composition and structure. The structured data can be exported in multiple formats. **c** Interactive scatter plot illustrating the relationship between the molar mass and the number of atoms in the ions within the database. **d** A hypothetical, unrelaxed Dion-Jacobson structure, generated using the pyrovskite package[61], employing the 2-(2-azaniumylethyldisulfanyl)ethylazanium conformer extracted from the database as the A-site cation as an example of the top layer of a 2D/3D perovskite heterostructure on top of a substrate with a transparent conductive oxide (TCO) and a hole transport layer (HTL).

Dion–Jacobson structure constructed using an A-site cation from the Perovskite Ion Database is shown in Fig. 3d. For a Jupyter Notebook demonstrating this workflow, see the code availability section.

To facilitate continued growth and completeness, the database allows new ions to be added if they are not already included in the curated dataset. Users encountering previously unconsidered perovskite ions are encouraged to contribute entries, further enhancing the value of this public resource. Instructions for how to add new ions are linked in the code and data availability section.

### Different types of perovskite samples

The composition of a perovskite film is often assumed to correspond to the stoichiometric ratio of the precursors used during synthesis. While this serves as a reasonable first approximation, it introduces some uncertainty in the reported composition. The perovskite composition can also be derived based on experimental data from spectroscopic or diffraction measurements. Since these approaches can yield different results, we propose that the basis for the reported composition – whether derived from precursor ratios, experimental measurements, or literature values – should be included in the description.

Beyond composition, the physical nature of the perovskite sample is also important, as it directly influences measurement outcomes and device integration. For example, due to quantum and dielectric confinement effects, bulk and nanosized perovskites can exhibit significantly different optoelectronic properties[34]. In the case of nanocrystals, the properties are further influenced by the morphology of the crystals[35]. To account for these variations, we recommend that the sample type, e.g. single crystal, polycrystalline thin film, colloidal solution, etc., as well as the perovskite dimensionality, i.e. 0D, 1D, 2D, or 3D, also should be specified as a part of the perovskite description.

### Additives, impurities, doping, and secondary phases

Additives are well known to influence key perovskite properties such as stability, defect concentration, fluorescence yield, etc.[36–42]. It would therefore be relevant to include the presence of additives within a description of the perovskite composition.

We propose that an additive should be defined as any substance deliberately introduced into the perovskite film, but which is not integrated into the perovskite crystal structure. However, the nature of additives does present some intriguing questions. In what form does an additive exist within the perovskite film? How to deal with substances for which it is unclear if they get integrated in the structure, form a benign secondary phase, or evaporate during the synthesis? An illustrative example of additives as benign phases is RbI. When RbI first was introduced as an additive[43], it was assumed that $Rb^+$ ions were incorporated on the A-site in the perovskite structure. However, subsequent research revealed that Rb instead forms a secondary, benign, non-perovskite phase[44,45]. Given this knowledge, Rb is thus now best categorized as an additive, rather than as an A-site cation which initially was the case. Another illustrative example is the role of chlorine ($Cl^-$) in $MAPbI_3$ perovskites. For a time, a popular recipe included an excess of methylammonium chloride, or $PbCl_2$, in the perovskite precursor solution[46]. The initial hypothesis was that most of the chloride disappeared into the gas phase during synthesis, but that a fraction of the $Cl^-$ made it into the perovskite's X-site[47,48]. The perovskite composition was therefore often referred to as $MAPbCl_xI_{3-x}$. However, subsequent studies showed that $Cl^-$ was not incorporated into the perovskite structure[49]. In light of this, $Cl^-$ could either be categorised as an additive, or only as part of one of the precursor chemicals featured in a detailed synthetic protocol.

Intentionally introduced dopants would be considered as a type of additive. Heterovalent metal doping has for example been explored as a strategy to tune electronic and luminescence properties[50–52].

Related to additives are impurities. The difference is that an impurity is a substance that is unintentionally present in the perovskite film. Impurities could result from for example residual solvents or unreacted precursors, uncontrolled byproducts from the synthesis of precursor chemicals, or from external contamination during processing.

To improve the quality of the perovskite description, we propose that it should include lists of additives and impurities present within the perovskite film. For each additive and impurity, we suggest including its name, the concentration or mass fraction, as well as additional data to uniquely identify the compound. To avoid the complexity related to large data schemas, we suggest encapsulating the details for additives and impurities within their own interoperable data schemas. These schemas would be very similar to the ones for the

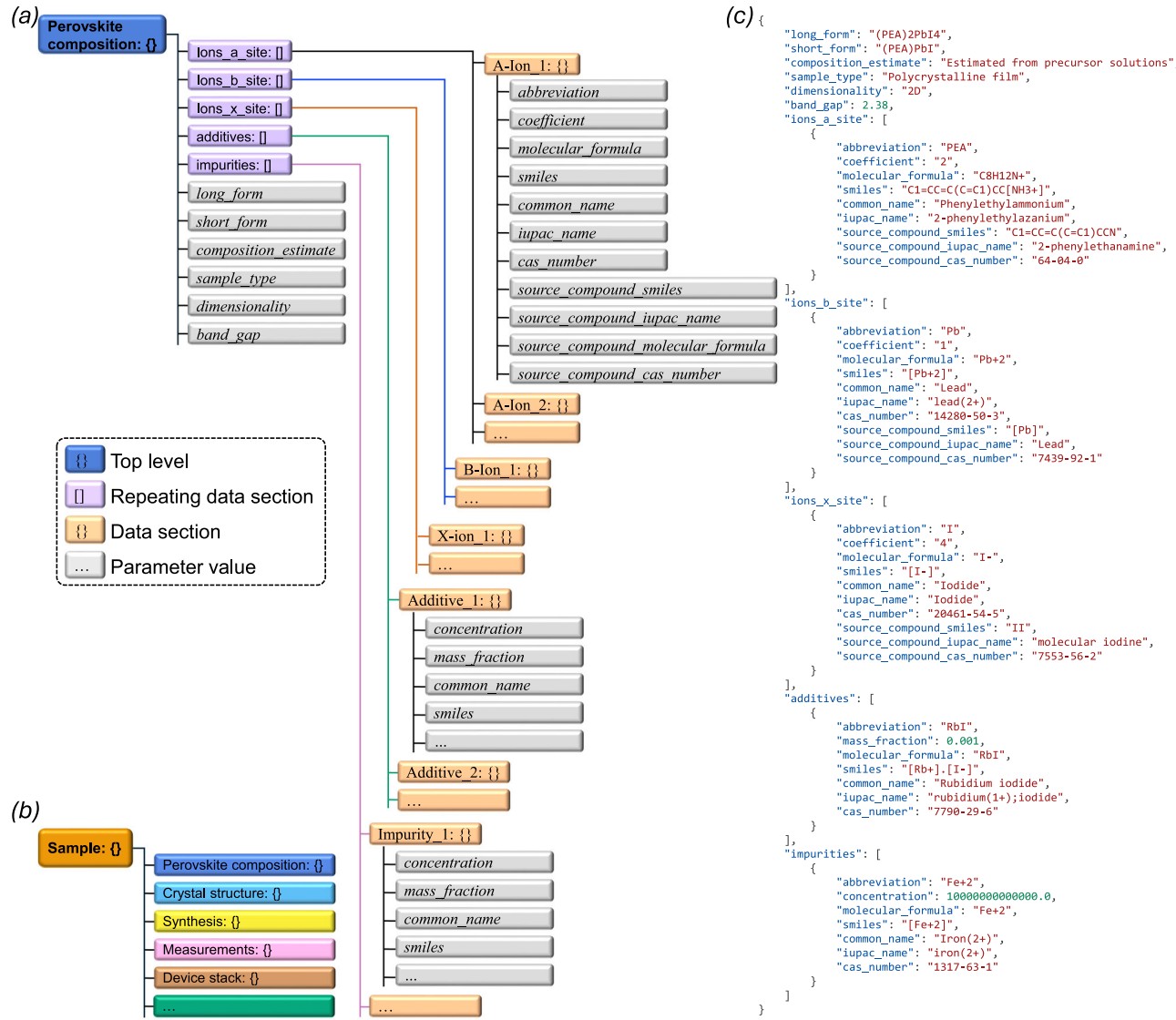

**Fig. 4 | Graphic summary of data schemas. a** The proposed schema for perovskite composition. **b** Illustration of how the proposed description of perovskite compositions could fit into a hierarchy of data schemas that together give a complete description of a sample. **c** Example JSON data for the 2D perovskite $(PEA)_2PbI_4$, with a RbI additive and a $Fe^{2+}$ impurity.

perovskite ions, and they could be integrated as individual, reusable, sub-modules within the data schema for the perovskite description.

## What to include and where to draw the line

A key consideration in data schema design is determining the appropriate level of detail to include. While more comprehensive descriptions are inherently more valuable, overly complex protocols tend to become cumbersome and less user-friendly, reducing their likelihood of being widely adopted. Based on insights from the Perovskite Database Project, we recommend a modular approach, where smaller, well-defined protocols are designed to capture specific aspects of a sample.

Figure 4a provides a graphical overview of the components we propose to be included in a structured description of perovskite compositions. In addition to the elements discussed earlier – i.e. sample type, perovskite dimensionality, ion-specific details, and lists of additives and impurities – we also include the perovskite's band gap. The motivation for this is the central importance the band gap has for optoelectronic applications. Band gaps are also routinely measured and reported. Although the band gap of a perovskite with nominally the same composition can vary slightly depending on the synthesis

protocol[15], it is largely an intrinsic property of the material, independent of device integration.

## Going beyond compositions towards descriptions of complete devices

A common experimental practice involves using a small number of perovskite compositions when fabricating a large number of devices. In such cases, the perovskite composition can be treated as a reusable data block that can be combined with other descriptive blocks. This approach enables the creation of a modular hierarchy of data classes that can be composed as needed to comprehensively describe a sample (Fig. 4b). The specific data blocks to include will depend on the sample type, the synthesis procedures, and the measurements being performed. For thin film samples, it would for example be natural to include a film morphology block with parameters specifying details such as the film thickness, average grains size, etc. Nanoparticle samples may instead include a block detailing the particle diameter, the particle size distribution, the nature of any capping ligands, etc. If structural data is available, the resulting CIF-file can be treated as an independent block.

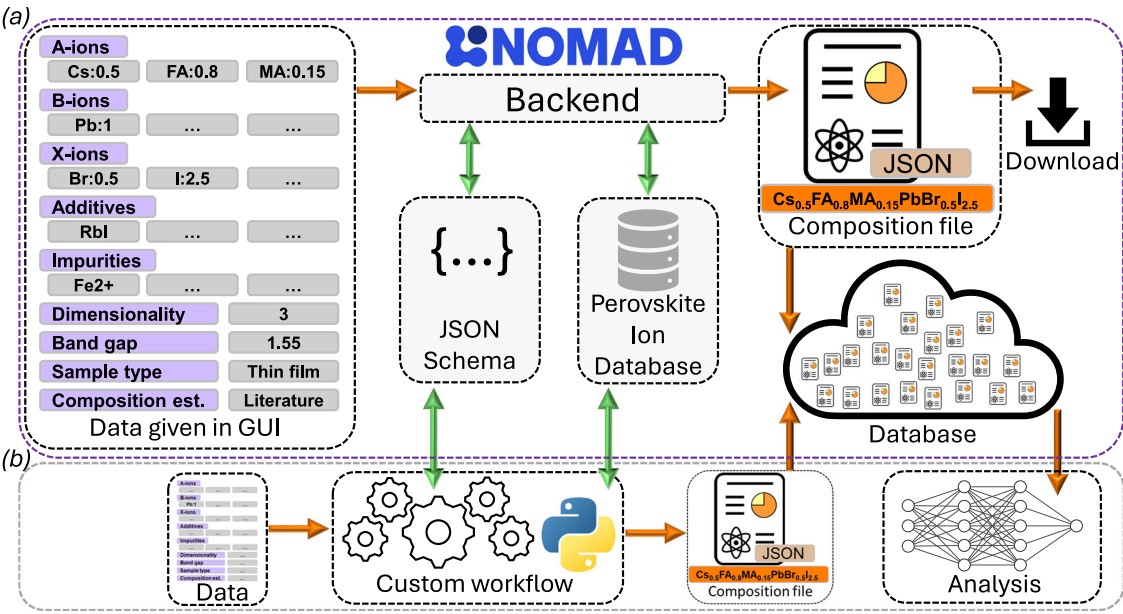

**Fig. 5 | Schematics of software utilities and workflows. a** Resources implemented in NOMAD. Includes a graphical user interface for data entry, the Hybrid Perovskite Ions Database, JSON schemas, a back end that generates validated JSON composition files, and online storage. **b** It is also possible to define custom workflows interacting with the NOMAD resources to accomplish the same thing.

The advantage of such a modular structure becomes evident when considering for example measurements, a device stack, or thin film deposition, where a large number of materials and synthetic procedures are in common use[15]. Each of these requires the specification of numerous parameters to ensure a complete and reproducible description. By treating these data schemas as independent yet interoperable units, it becomes possible to efficiently assemble detailed sample descriptions while avoiding unnecessary duplication of effort.

In this paper, the focus is limited to defining a standardised schema for perovskite compositions, but we plan to extend this approach in future work by developing additional modules and strategies for a more comprehensive description of complete perovskite devices.

## JSON schema for perovskite compositions

To integrate the ideas discussed above into a structured, interoperable format, we propose using JSON files. JSON (JavaScript Object Notation) is an open standard widely adopted for data storage, transmission, and dissemination through name-value pairs. It is compatible with modern programming languages while remaining human-readable, making it an effective and practical choice for representing perovskite composition data[53].

A detailed description of the proposed JSON Schema, including all key-value pairs, is provided in the Supporting Information. Figure 4c illustrates what a typical perovskite composition JSON file could look like. To ensure consistency, we have developed a JSON Schema for validation, available in both the Supporting Information and the project's GitHub repository.

Once a perovskite composition is structured in this way, the resulting JSON file can be reused consistently. When publishing experimental work, including such a file in the Supporting Information provides a concise yet comprehensive and unambiguous description of the perovskite composition. An even better approach would be to also deposit the JSON file in an online repository with a persistent link. Beyond standardized data sharing, this would enable programmatic access to the file which would simplify reproduction of results as well as large-scale data aggregation and machine learning applications.

## Tools for generating and sharing composition files

To facilitate the creation of perovskite composition files according to the guidelines specified here, we have developed a set of software tools (Fig. 5 and Supplementary Figs. 1, 2). The primary tool is an editor built into NOMAD[54] that allows users to create composition files in the graphical user interface based on the schema definition. A detailed description and instructions are linked to in the SI section.

The NOMAD interface is directly linked to the Hybrid Perovskite Ions Database[31]. This ensures consistency and minimizes user input by eliminating the need to manually define ion details every time a composition file is generated. When a perovskite is composed of only previously reported ions, the user only needs to specify standard ion abbreviations, stoichiometric coefficients, and descriptive parameters (i.e. dimensionality, band gap, composition estimate, and sample type). A normalization algorithm then generates a standardised and validated JSON file following the proposed schema. If any ions are not yet defined in the database, they can be added beforehand as described earlier.

The generated composition files can be directly downloaded. They can also be seamlessly uploaded to NOMAD, where they can be compiled into a dataset, assigned a DOI, and referenced in related works. Once stored in NOMAD, the composition files become programmatically accessible via the NOMAD API, which enables seamless integration into data analysis and machine learning applications.

Although the NOMAD interface provides a user-friendly and efficient tool for generating and disseminating perovskite compositions, the JSON Schema is platform-agnostic. This allows users to integrate the schema into their own workflows, structure data mining from literature, use it in research data management frameworks, or use it to manually construct and validate composition and ion files. Regardless of what tools and workflows are being used, if perovskite composition files are consistently created, disseminated, and made accessible upon publication, it would represent a significant step forward in terms of FAIR data management.

As a complement to the NOMAD implementation, and for exemplifying an alternative workflow, we have developed a set of Python utilities. Those can be integrated into custom workflows for generating composition files (Fig. 5b). We have also developed a minimalistic

Python-based GUI for constructing perovskite composition files (see Supplementary Fig. 2 and code resources for examples). These tools perform input validation, normalization, and automatic formatting, ensuring compliance with the JSON Schema. The Python utilities, GUI, instructions, and an example Jupyter notebook demonstrating the functionality can be found in the projects GitHub repository (see the SI section).

### Edge cases, best practises, and protocol extensibility

No matter how a data schema is constructed, there will always be edge cases that do not fit neatly within the structure, or which could be categorized in multiple ways.

An example of an edge case is when a 3D perovskite film undergoes a surface treatment that converts its topmost layer into a capping 2D perovskite layer. Should this be treated as a single perovskite with a secondary phase, or as two distinct perovskites? In this scenario, we recommend treating the sample as a layered structure and generating separate composition files for the two phases.

A similar edge case occurs when a perovskite sample contains two distinct phases, such as when a 2D phase is intermixed with a 3D phase. Here again, our recommendation is to generate individual data files for each phase and describe the sample as a mixture of the two.

A JSON Schema defenition is inherently flexible and can be updated as new parameters become relevant. No schema is perfect, and we anticipate a future desire to modify and extend the schema here presented. For instance, someone may wish to include more key-value pairs to capture additional relevant parameters measured in the lab and included in custom workflows. While such extensions could introduce challenges related to backward compatibility, schema versioning, etc., a good pragmatic guiding principle is to not make the perfect the enemy of the good. In practice, an extended JSON file containing additional data remains processable by workflows based on the current schema.

### The role of publishers, journals, and editors

For the proposed data schema to have an impact beyond improving individual publications and the data management of groups that adopt it, it must be widely adopted. Only then can it serve as an effective tool for data mining, machine learning, and large-scale comparative analysis.

One potential pathway towards widespread adoption would be if editors and major publishing houses were to embrace and endorse this standard, or something similar in spirit. If journals incentivise and actively encourage authors to use a structured and formalised approach to reporting perovskite composition data during the peer-review process, they could play a key role in establishing new norms for increased data transparency and reproducibility.

There are existing precedents for such editorial requirements. Examples include the demand for using public repositories to disclose new gene sequences and for sharing CIF files when reporting new crystal structures. There are also precedents within the perovskite community. Journals such as Nature and Energy & Environmental Science are for example demanding authors to go through checklists before publishing perovskite solar cell device data[55,56]. These examples demonstrate that protocols for structured data sharing can be successfully adapted and developed into standard practice, at least if we can muster the collective will to enforce them. However, we acknowledge that the sheer number of journals and fragmentation of publishing standards poses a problem for an exclusive top-down approach. A widespread schema adoption will likely require a combination of a bottom-up cultural change and an external editorial pressure. By providing an open schema, examples, and digital utilities, we hereby provide tools that could spark such a combination.

### Summary

The number of hybrid perovskites explored to date is large and growing. We have identified nearly 300 different A-site cations that have been used in hybrid perovskites. Large is also the variability in how to write and communicate perovskite compositions, leading to confusion and lack of interoperability. To address this issue, we propose a set of guidelines and a structured data model based on a JSON schema to standardise the description and reporting of perovskite compositions. This schema adheres to IUPAC recommendations and includes molecular formulas, SMILES strings, IUPAC names, and CAS numbers for all ions and their neutral parent compounds. By adopting this structured approach, we would eliminate the ambiguity concerning perovskite compositions that plagues the scientific literature and achieve greater consistency, reproducibility, and interoperability. The proposed schema is human-readable and machine-compatible, possible to reuse and expand, and provides a robust foundation for future data analysis, modelling, and computational projects. To facilitate adoption, we provide a graphical user interface implemented in NOMAD, a database of all perovskite ions with associated data, and Python utilities. These tools facilitate the generation of structured, comprehensive, and standardized descriptions of perovskite compositions from ion abbreviations and stoichiometric coefficients. A platform to publish the generated data in NOMAD is also provided. Describing a perovskite in this way does not require much effort. If widely adopted, this approach would eliminate ambiguities in reported perovskite compositions as well as enable better utilization of the growing body of perovskite data for machine learning and computational studies.

## Methods

The perovskite ions have bene identified and curated from the Perovskite Database[15], the 2D-perovskite database[19], and the product catalogue from Great Cell Solar. The additional data for the perovskite ions have been extracted from PubChem.

In the Halide Perovskite Ions Database, the conformers of the A-site cations were generated by converting each ion's SMILES string into a three-dimensional structure using RDKit[28]. First, the SMILES representation is parsed to create an RDKit molecule object, and explicit hydrogens are added to ensure a complete atomic representation. Next, a 3D geometry is generated using RDKit's embedding algorithm, which utilizes distance geometry to position the atoms in space. The initial conformer is then refined by performing an energy minimization with the Merck Molecular Force Field (MMFF)[57], thereby optimizing the molecular geometry.

## Data availability

A compilation of all identified perovskite ions complemented with associated data are provided in: a) an Excel file found in the supporting information, b) The projects GitHub repository: https://github.com/Jesperkemist/Perovskite_composition, c) The Hybrid Perovskite Ions Database hosted by NOMAD, found at: https://nomad-lab.eu/prod/v1/gui/search/perovskite-ions.

## Code availability

The code for the NOMAD resources is available at: https://github.com/FAIRmat-NFDI/nomad-perovskite-solar-cells-database (release 1.2.0). The code is also released at Zenodo at https://zenodo.org/records/16895005[58]. The code for generating the perovskite in Fig. 3.d, and code showing how to query the database for perovskite ions and perovskite compositions are available at: https://github.com/FAIRmat-NFDI/nomad-perovskite-solar-cells-database/tree/main/src/perovskite_solar_cell_database/example_uploads/ions_database The Python utilities for generating composition files, accessing and manipulating perovskite compositions and ions stored in the NOMAD

database, and the Python-based GUI is found at: https://github.com/Jesperkemist/Perovskite_composition. The code is also released at Zenodo at https://doi.org/10.5281/zenodo.16895005[59].

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

## Acknowledgements

T.J.J. would like to acknowledge Åforsk (Grant No. 23-629), the Advance Center of Functional Materials (AFM) at Linköping University (Grant No. 310437), Carl Tryggers Stiftelse (Grant No. CTS 24: 3375), the Ministry of Science and Technology in China via the National Key Research and Development Program of China (Grant No. 2021YFF0500501), and Applied Basic Research Projects in Tianjin, (Grant No. 22JCYBJC01530) for financial support. H.H., C.D., and J.A.M. are part of the NFDI consortium FAIRmat funded by the Deutsche Forschungsgemeinschaft (DFG, German Research Foundation) – project 460197019, and of the SolMates project, funded by the European Union's Horizon Europe research and innovation program under grant agreement No 101122288. R.C. and V.B. were supported by the National Science Foundation under Award Number 2323803.

## Author contributions

T.J.J. come up with the idea, have supervised data gathering, and been responsible for writing the manuscript. A.M., C.C., L.Q., and T.J.J. have composed the list of perovskite ions and gathered the data for the perovskite ions database. H.N., J.A.M. and T.J.J have written the code. H.N. and J.A.M. have been responsible for integrating the resources in NOMAD. A.M., C.C., L.Q., J.L., H.N. R.C., V.B. E.U., C.D., J.A.M., and T.J.J. have all been involved in discussions, designing the data schemas, and reviewing the manuscript.

## Funding

## Competing interests

The authors declare no competing interests.
