## [Transparent Peer Review file · Nature Communications]

Towards an Interoperable Perovskite Description or How to Keep Track of 300 Perovskite Ions

Corresponding Author: Dr T. Jesper Jacobsson

Version 1:

Reviewer comments:

Reviewer #1

(Remarks to the Author)

This manuscript presents a standardized, machine-readable data schema for describing hybrid organic-inorganic perovskite compositions. Motivated by the lack of consistency in how perovskite materials are reported in the literature, the authors develop a JSON-based schema that includes canonical identifiers for each ion (such as SMILES, IUPAC name, CAS number), and define best practices for reporting sample composition, dimensionality, additives, and impurities. A curated database of nearly 300 A-site cations is provided, along with graphical and programmatic tools integrated into the NOMAD platform, facilitating the creation and sharing of interoperable composition files. The proposed modular schema is thoughtfully designed, technically robust, and aligns with FAIR data principles. It has the potential to improve reproducibility and data reuse in the perovskite community, especially for researchers working on data mining, machine learning, or automated literature analysis.

However, there are a few concerns about this work:

1. While the infrastructure presented is valuable, the manuscript does not provide new physical insights into perovskite materials, nor does it report experimental or computational results with scientific implications. The work is methodological and infrastructural in nature.
2. The database focuses solely on ion identity and composition, without linking to physical properties (e.g., band gaps, mobilities, stability metrics). As a result, its utility for materials discovery or understanding structure–property relationships is limited.
3. The impact of the work lies in improving data handling and metadata curation, which, while valuable, is not aligned with the conceptual or mechanistic advances typically expected in Nature Communications.

I commend the authors for the technical quality of their work and the effort to promote FAIR practices in the perovskite research community. However, given the primarily infrastructural nature of the manuscript and the lack of new scientific insight or conceptual innovation, I believe this work is better suited for publication in a specialized journal focused on materials informatics, data infrastructure, or research data management, such as Scientific Data or Patterns.

(Remarks on code availability)

Reviewer #2

(Remarks to the Author)

This is a timely contribution to the perovskite community. The authors present a practical framework for standardising hybrid perovskite composition data, supported by intuitive tools and a rich, curated database. Figure 2 alone underscores the scale of the problem and the thoughtful effort invested in resolving it. The proposed schema and utilities are well-aligned with FAIR principles and will facilitate data sharing and machine learning applications.

I can recommend this work for publication in Nature Communications, but I do have two very minor suggestions.

- Remove the phrase “mind, body, and soul” from the acknowledgements, as it distracts from the otherwise professional tone.

- The section “The Role of Publishers, Journals, and Editors” might underappreciate the challenges faced by researchers and PIs. Bottom-up cultural change and community training could be equally important levers. Consider acknowledging that the sheer number of journals and fragmentation of publishing standards may hinder widespread schema adoption, systematic change will require incentives beyond editorial mandates. Emphasise that the schema is freely available and examples are available to lower the barriers to usage by the perovskite community.

(Remarks on code availability)

The codebase looks good, but I suggest a persistent DOI for future proofing, e.g.

<https://docs.github.com/en/repositories/archiving-a-github-repository/referencing-and-citing-content>

Reviewer #3

(Remarks to the Author)

This manuscript presents a valuable technical resource for the perovskite research community: a standardized schema and supporting utilities for the structured description and reporting of hybrid perovskite compositions. The work addresses a well-recognized issue in the field—namely, the inconsistency and ambiguity in perovskite nomenclature that hinders data sharing, mining, and machine learning applications. The provision of a curated database and automated tools enhances the practical utility of this effort.

However, while the infrastructure developed is well-executed and likely to be useful to researchers working in perovskite materials and optoelectronics, the manuscript does not provide new scientific insights or methodological advances. The work primarily constitutes a data and software standardization effort, rather than a conceptual or theoretical contribution. As such, it falls outside the scope of what Nature Communications typically seeks to publish, which is original research that advances scientific understanding or methodology.

I therefore recommend rejection of the manuscript in its current form. That said, I believe the work would be highly suitable for publication in a specialized journal focused on materials informatics, scientific data infrastructure, or community data standards—such as Scientific Data, npj Computational Materials, or Data in Brief, where it could have significant impact and visibility within the relevant user communities.

(Remarks on code availability)

Response to reviewers' comments

Answers to Reviewer's comment

Reviewer 1

This manuscript presents a standardized, machine-readable data schema for describing hybrid organic-inorganic perovskite compositions. Motivated by the lack of consistency in how perovskite materials are reported in the literature, the authors develop a JSON-based schema that includes canonical identifiers for each ion (such as SMILES, IUPAC name, CAS number), and define best practices for reporting sample composition, dimensionality, additives, and impurities. A curated database of nearly 300 A-site cations is provided, along with graphical and programmatic tools integrated into the NOMAD platform, facilitating the creation and sharing of interoperable composition files. The proposed modular schema is thoughtfully designed, technically robust, and aligns with FAIR data principles. It has the potential to improve reproducibility and data reuse in the perovskite community, especially for researchers working on data mining, machine learning, or automated literature analysis.

However, there are a few concerns about this work:

Comment

1. While the infrastructure presented is valuable, the manuscript does not provide new physical insights into perovskite materials, nor does it report experimental or computational results with scientific implications. The work is methodological and infrastructural in nature.

Answer

That is basically a correct observation.

Comment

2. The database focuses solely on ion identity and composition, without linking to physical properties (e.g., band gaps, mobilities, stability metrics). As a result, its utility for materials discovery or understanding structure–property relationships is limited.

Answer

We agree that having more data, like physical properties, linked to the composition would make for a more valuable resource. However, as we argue in the subsection “*What to Include and Where to Draw the Line*”, if we are going to achieve that, the best approach is probably to go towards a hierarchy of modular protocols, where one would be for compositions and one for physical properties, and one for synthesis, etc. We share in what we think is the vision of the reviewer, and we think that what we present here is one important, self-contained stepping stone towards that goal.

Comment

3. The impact of the work lies in improving data handling and metadata curation, which, while valuable, is not aligned with the conceptual or mechanistic advances typically expected in Nature Communications.

I commend the authors for the technical quality of their work and the effort to promote FAIR practices in the perovskite research community. However, given the primarily infrastructural nature of the manuscript and the lack of new scientific insight or conceptual innovation, I believe this work is better suited for publication in a specialized journal focused on materials informatics, data infrastructure, or research data management, such as Scientific Data or Patterns.

Answer

We are glad to see that the reviewer appreciates this work. Concerning whatever or not Nature Communications is the right journal to publish it. We think it is, but in the end that is an editorial decision.

Reviewer 2

Comment

This is a timely contribution to the perovskite community. The authors present a practical framework for standardising hybrid perovskite composition data, supported by intuitive tools and a rich, curated database. Figure 2 alone underscores the scale of the problem and the thoughtful effort invested in resolving it. The proposed schema and utilities are well-aligned with FAIR principles and will facilitate data sharing and machine learning applications.

I can recommend this work for publication in Nature Communications, but I do have two very minor suggestions.

Answer

We are happy to hear that the reviewer liked our work.

Comment

- Remove the phrase “mind, body, and soul” from the acknowledgements, as it distracts from the otherwise professional tone.

Answer

Ok. We have now removed that somewhat more flowery sentence.

Comment

- The section “The Role of Publishers, Journals, and Editors” might underappreciate the challenges faced by researchers and PIs. Bottom-up cultural change and community training could be equally important levers. Consider acknowledging that the sheer number of journals and fragmentation of publishing standards may hinder widespread schema adoption, systematic change will require incentives beyond editorial mandates. Emphasise that the schema is freely available and examples are available to lower the barriers to usage by the perovskite community.

Answer

We agree.

We have now in the section “The Role of Publishers, Journals, and Editors” added the following paragraph to clarify this:

However, we acknowledge that the sheer number of journals and fragmentation of publishing standards poses a problem for an exclusive top-down approach. A widespread schema adoption will likely require a combination of a bottom-up cultural change and an external editorial pressure. By providing an open schema, examples, and digital utilities, we hereby provide the tools that can spark such a combination.

Comment

The codebase looks good, but I suggest a persistent DOI for future proofing, e.g.

<https://docs.github.com/en/repositories/archiving-a-github-repository/referencing-and-citing-content>

Answer

We have now complemented the code with a Zenodo release.

Reviewer 3

Comment

This manuscript presents a valuable technical resource for the perovskite research community: a standardized schema and supporting utilities for the structured description and reporting of hybrid perovskite compositions. The work addresses a well-recognized issue in the field—namely, the inconsistency and ambiguity in perovskite nomenclature that hinders data sharing, mining, and machine learning applications. The provision of a curated database and automated tools enhances the practical utility of this effort.

However, while the infrastructure developed is well-executed and likely to be useful to researchers working in perovskite materials and optoelectronics, the manuscript does not provide new scientific insights or methodological advances. The work primarily constitutes a data and software standardization effort, rather than a conceptual or theoretical contribution. As such, it falls outside the scope of what Nature Communications typically seeks to publish, which is original research that advances scientific understanding or methodology.

I therefore recommend rejection of the manuscript in its current form. That said, I believe the work would be highly suitable for publication in a specialized journal focused on materials informatics, scientific data infrastructure, or community data standards—such as Scientific Data, npj Computational Materials, or Data in Brief, where it could have significant impact and visibility within the relevant user communities.

Answer

We notice that the reviewer seems to fundamentally like our work and see the value in it. We appreciate that. The only concern seems to be whatever Nature Communication is the right journal for our work. We think it is, but that is in the end an editorial decision.